# Review of Rhinitis: Classification, Types, Pathophysiology

**DOI:** 10.3390/jcm10143183

**Published:** 2021-07-19

**Authors:** Georgia A. Liva, Alexander D. Karatzanis, Emmamuel P. Prokopakis

**Affiliations:** Department of Otorhinolaryngology, Medical School, University of Crete, 71500 Heraklion, Crete, Greece; georgialiva21@gmail.com (G.A.L.); akaratzanis@yahoo.com (A.D.K.)

**Keywords:** rhinitis, review, allergic, non-allergic, occupational, gustatory, idiopathic, atrophic

## Abstract

Rhinitis describes a pattern of symptoms as a result of nasal inflammation and/or dysfunction of the nasal mucosa. It is an umbrella entity that includes many different subtypes, several of which escape of complete characterization. Rhinitis is considered as a pathologic condition with considerable morbidity and financial burden on health care systems worldwide. Its economic impact is further emphasized by the fact that it represents a risk factor for other conditions such as sinusitis, asthma, learning disabilities, behavioral changes, and psychological impairment. Rhinitis may be associated with many etiologic triggers such as infections, immediate-type allergic responses, inhaled irritants, medications, hormonal disturbances, and neural system dysfunction. It is basically classified into three major clinical phenotypes: allergic rhinitis (AR), infectious rhinitis, and non-allergic, non-infectious rhinitis (NAR). However, this subdivision may be considered as an oversimplification because a combined (mixed) phenotype exists in many individuals and different endotypes of rhinitis subgroups are overlapping. Due to the variety of pathophysiologic mechanisms (endotypes) and clinical symptoms (phenotypes), it is difficult to develop clear guidelines for diagnosis and treatment. This study aims to review the types of allergic and non-allergic rhinitis, providing a thorough analysis of the pathophysiological background, diagnostic approach, and main treatment options.

## 1. Introduction

Rhinitis is an entity that includes many different subtypes and is mainly used to describe a pattern of nasal symptoms such as nasal congestion/obstruction, rhinorrhea, sneezing and pruritus that appear as a result of inflammation and/or dysfunction of the nasal mucosa [1,2,3,4]. There are three distinct rhinitis subgroups that are widely accepted: allergic rhinitis (AR), infectious rhinitis, and non-allergic, non-infectious rhinitis (NAR) [5]. These phenotypes, however, are dynamic and may develop into one another. Therefore, caution against oversimplification should be advised since an overlapping or combined phenotype may exist in several patients [5,6]. For phenotype classification, various criteria may be used, including the severity of disease (mild, moderate/severe), pattern of symptoms (seasonal/perennial or intermittent/persistent), predominant symptom (runners/blockers), possible triggering factor (allergens, infectious agents, etc.) and response to treatment (controlled/uncontrolled) [7,8,9,10]. Recently, another disease categorization has been proposed based on endotype, and grouping rhinitis depending on the specific pathophysiological pathway [1,5].

The prevalence of allergic rhinitis in the United States of America ranges from 9% to 42%, which is translated to approximately 58-million people, when the prevalence of non-allergic rhinitis appears to be 19 million and mixed rhinitis 26-million people [4]. In the UK, the prevalence reaches 26% in adults, with an observed peak in the third and fourth decades of age [11,12]. Rhinitis is considered one of the most common medical conditions, with significant impairment of quality of life. Apart from upper airway symptoms, sleeping and psychological disturbances, decreased work productivity and school performance impairment must be taken under consideration [2,3]. Rhinitis is also associated with a considerable financial burden [1]. All of the above constitute the indirect costs of rhinitis, but there also exist direct costs such as physician office visits, lab tests and medication [1,4].

## 2. Allergic Rhinitis

Allergic rhinitis is a well-defined endotype according to ARIA. It is defined as an IgE-mediated, type 1 hypersensitivity response to a spectrum of inhaled environmental allergens [5,13,14,15]. Allergic rhinitis is characterized by anterior or posterior rhinorrhea, nasal congestion/blockage, itching of the nose, and sneezing occurring for more than one hour on two or more consecutive days [8]. According to ARIA, allergic rhinitis is categorized based on symptom duration, intermittent and persistent, and severity, mild–moderate and severe. Allergens associated with allergic rhinitis are proteins that come from airborne particles including pollens, dust mites, insect feces, animal dander, and molds [5,13,14,15]. The common comorbidities associated with allergic rhinitis are asthma and conjunctivitis. Its strong correlation with asthma may be explained by the theory of the unified airway, which dictates that the upper and lower airway inflammation share common pathophysiologic mechanisms, coexisting and communicating via the systemic circulation [16,17,18,19]. Clinical expression of the disease is a result of a cascade of immunological and biochemical events. Allergens are inhaled, superimposed to nasal mucous, and diffuse into nasal tissues. Then, antigen-presenting cells (APCs) break antigens into antigenic peptides and migrate to lymph nodes to present the peptides to naïve CD4+ T lymphocytes (T cells) [13].

The process of activation of CD4+ T lymphocytes includes the interaction of specific surface T-cell receptors with allergen MHC class II complexes on the APCs [13,20]. Dendritic Cells (DCs) and signals from antigen presentation assist the differentiation of naïve T helper cells to Th1 or Th2. Th2 lymphocytes activate the production of specific cytokines which cause the synthesis of IgE antibodies from B-cells. IgE antibodies have the ability to bind to high-affinity receptors on the surface of dendritic cells, on low-affinity receptors on monocytes-macrophages and B-lymphocytes and on high-affinity tetrameric receptors FcεRl on mast cells and on basophils [13,21,22]. The latter interaction induces the cellular allergic reaction and the activation of several signaling cascades. One of these leads to granule exocytosis and release of preformed or newly created inflammatory mediators (such as histamine, leukotrienes, prostaglandins, platelet-activating factor, etc.).

The nasal allergic reaction is distinguished in early and late phases. The symptoms of early phase begin almost immediately after exposure to the responsible allergen, arrive at a peak in a few minutes, and subside within one to several hours [1,5]. Within minutes from the exposure, the interaction between IgE and allergen leads to degranulation of mast cells and release of inflammatory mediators such as leukotrienes, prostaglandins, cytokines, and histamine. These molecules are responsible for symptoms such as sneezing, itching, rhinorrhea, and nasal congestion [1,23]. Histamine binds on the H1 receptors and provokes virtually all of the early phase symptoms. During the late phase, the most dominant symptom is nasal congestion. The release of mediators that has taken place in the early phase leads to infiltration of nasal mucosa by basophils, eosinophils, neutrophils, mast cells and mononuclear cells. The mast cells have been found to play a prominent role not only in the allergic response but also in sustaining the allergic response chronically. This is mainly related to the fact that mediators produced by the degranulation of mast cells and histamine play an important role in the recruitment of Th2 lymphocytes to target organs [23,24]. The cysteinyl leukotrienes are mainly responsible for the activation of eosinophils. Eosinophils are predominant in the late phase response and are associated with the progression of allergic symptoms [23]. Proinflammatory mediators, cationic proteins, eosinophil peroxidase, and cysteinyl leukotrienes are released from eosinophils [1,24].

A thorough medical history and a detailed clinical examination may lead to the suspicion of allergic rhinitis. The diagnostic tests for allergic rhinitis are separated into in vivo, which are percutaneous skin tests (skin prick tests); and in vivo, including the radioallergosorbent test (RAST), multiple antigen simultaneous testing (MAST), fluoroallergosorbent test (FAST), and immunoassay capture test (ImmunoCAP). The most common diagnostic tests for allergic rhinitis are percutaneous skin test (skin prick test) and the allergen-specific immunoglobulin E (IgE) antibody test (RAST), which recently has been replaced with ImmunoCAP tests. Skin prick testing involves introducing controlled amounts of allergen and control substances into the skin. Skin testing provokes both types of responses, early and late; however, the main goal is detecting the immediate allergic response caused by the release of mast cell or basophil IgE-specific mediators, which create the classic wheal-and-flare reaction within fifteen minutes [25]. A positive result is defined as a wheal ≥3 mm diameter [25]. Allergen-specific IgE antibody testing (radioallergosorbent testing [RAST]) is useful in primary care if percutaneous testing is not practical (e.g., problems with reagent storage, expertise, frequency of use, staff training) or if it is contraindicated (e.g., medication such as tricyclic antidepressants, antihistamines) [1,25]. Even if RAST is highly specific, it is costly, time-consuming as to the results and not as sensitive as skin testing. Although the available commercial RAST products are generally reliable, they do not always provide reproducible, accurate data. The ImmunoCAP system is an in vitro test using three-dimensional cellulose solid allergen phase in order to detect specific IgE to allergen components. It has been found that ImmunoCAP tests have similar sensitivity for house dust and lower sensitivity for pollen, dog dander and Candida compared to skin prick tests [26]. The MAST test uses no radioactive agents and allows simultaneous examination of multiple antigens. The MAST system provides similar information as the CAP system [26,27,28]. However, the CAP system seems to have better sensitivity [28]. The FAST is a method that measures specific serum IgE by a chemical-radiating method such as MAST and was first used as an inhibition assay in order to determine cross-reaction activity between aeroallergens [26,29]. In comparison with MAST, FAST requires less time to be analyzed and less serum quantity to be used [26,29].

Medical therapy includes intranasal corticosteroids, which are safe in administration to adults and children and are superior to the combination of oral antihistamines and leukotriene receptor antagonists (LTRAs) [1,30]. First-generation antihistamines are no longer recommended due to their side effects, while second generation oral antihistamines have strong H1 receptor selectivity and weak anticholinergic action. Intranasal corticosteroids show efficacy in controlling allergic rhinitis symptoms and are found to be more effective than intranasal antihistamines. The combination of intranasal corticosteroids and intranasal antihistamines has been shown to be even more effective than each agent alone [1,31,32]. Nasal irrigation is widely used in all types of rhinitis with isotonic or hypertonic saline and helps in the removal of mucous and the clearance of inflammatory medication. It remains unclear if hypertonic saline has a better effect compared to isotonic saline [33]. Intranasal cromolyn/nedocromil are used prophylactically in AR because of their inhibitory activity on mast cells degranulation through stabilizing the membrane of mast cells [1,34]. Even though their use is safe, it seems that their action is less effective compared to topical corticosteroids/antihistamines because they have no action in already released inflammatory factors [34]. Omalizumab is an anti-IgE humanized monoclonal antibody which is approved for chronic urticaria and severe allergic asthma [35]. It has been found that it reduces nasal and ocular symptoms in AR but has not been approved for treatment of AR yet [1,36]. Specifically, a randomized placebo control, double blinded trial was conducted which proved that omalizumab prevents and controls nasal and ocular symptoms in moderate–severe seasonal allergic rhinitis to pollen in Japan [37]. However, it has not been approved for seasonal allergic rhinitis yet [37]. Mepolizumab, reslizumab and benralizumab are humanized mAbs against Il-5 which have an effect on eosinophilic asthma; the effects are even promising in AR, and are still under further investigation [1,38].

Individuals with allergic rhinitis (AR) are positive for the skin prick test and/or serum-specific (s)IgE/RAST. However, some patients with seasonal or perennial rhinitis have negative SPT and RAST and a positive nasal provocation test (NAPT) for specific allergens. This phenotype has been termed local allergic rhinitis (LAR) and is not included in AR or NAR groups [36,38,39]. LAR is characterized by nasal mucosa localized allergic response type 2 [19] with the presence of nasal-specific IgE (NsIgE). The typical profile of patients with LAR includes mostly young women, non-smokers with moderate/severe rhinitis, with persistent/perennial clinical behavior, and with conjunctivitis and asthma [38]. House dust mites are the most common allergic triggers in LAR. Apart from mites, the mold alternaria is an allergen more often found in LAR, while pollen and animal dander appear more often in patients with AR [19,38]. The diagnosis of LAR begins with clinical history, family history of atopy/asthma, and exclusion of CRS with nasal endoscopy and/or CT scan [38]. If the SPT and sIgE are positive, the diagnosis of AR is made. In case these are negative, the response of the target organ to an allergen challenge must be evaluated [19]. The gold standard in LAR diagnosis is the nasal allergy provocation test (NAPT), and alternatively the detection of sIgE in nasal secretions or a positive basophil activation test (BAT) [19]. It is worth mentioning that there are patients with persistent symptoms of rhinitis who are positive only to seasonal allergens on SPTs. Some of these patients are positive to NAPT to perennial allergens with their rhinitis phenotype characterized as dual allergic rhinitis (DAR), referring to the contemporary local and systemic sensitization in the same individual [38].

## 3. Non-Allergic Rhinitis (NAR)

Non-allergic rhinitis is a chronic rhinitis without clinical manifestations of endonasal infection and systemic allergic inflammation (negative SPT, negative total blood IgE, and RAST tests) [1,5]. Non-allergic rhinitis represents a heterogenous group of patients which may be classified into at least six subgroups:(1)Drug-induced rhinitis(2)Hormone-induced rhinitis(3)Senile rhinitis or rhinitis of the elderly(4)Gustatory rhinitis(5)Occupational rhinitis(6)Idiopathic rhinitis(7)Atrophic rhinitis

Further classification of NAR has been proposed, based on the cellular inflammatory profile. NARES (non-allergic rhinitis with eosinophilia) and neutrophilic NAR are the most common types, with NARES defined by more than 20% eosinophils in nasal smears without any evidence of allergy or other nasal pathology and is associated with other comorbidities such as asthma [5,40]. However, as to whether NARES represents a distinct phenotype or its pathophysiologic mechanisms overlap with various other conditions is controversial [5]. Neutrophilic NAR is defined by infiltration of equal to or more than 20% neutrophils on the nasal smear without the presence of other inflammatory organisms such as bacteria or fungi [40]. NAR with mast cells and mixed NAR (with eosinophils and mast cells) are less distinguishable, less common, and more difficult to treat [40].

### 3.1. Drug-Induced Rhinitis

Drug-induced rhinitis is classified into three subgroups: (a) local inflammatory type, (b) neurogenic type, and (c) idiopathic type.

Local inflammatory type: aspirin and non-steroidal anti-inflammatory drugs (NSAIDs) are drugs that may cause an acute nasal inflammatory response with the inhibition of cyclooxygenase 1 (COX-1) mechanism. Inhibition of COX-1 increases the metabolism of arachidonic acid due to the lypoxygenase pathway, resulting in a decrease of prostaglandin 2 (PGE2). The decrease of PGE2 is followed by an elevation of cysteinyl leukotrienes; LT4, D4, E4, with LTC4 characterized as the cysteinyl leukotriene responsible for aspirin-exacerbated asthma [1,41]. Cysteinyl leukotrienes may cause bronchoconstriction, mucosal edema and hypersecretion, and vasoconstriction. They are responsible for the release of chemotaxis of eosinophils in human airways [41,42]. The cysteinyl leukotrienes are activated after their interaction with cysteinyl –LT receptors. It has been shown that the responsible enzyme for cysteinyl LT activation is LTC4 synthase, which is overexpressed in aspirin-induced rhinitis [42].

Neurogenic type: the airways are innervated by sympathetic, parasympathetic, and sensory fibers, which are located near nasal mucosa blood vessels and secretory glands [41]. The neurogenic type occurs with sympatholytic drugs such as a and b adrenergic antagonists, which cause down-regulation of the sympathetic tone, resulting in vascular dilation, nasal congestion, and rhinorrhea. Phosphodiesterase-5 selective inhibitors such as sildenafil, tadalafil, and vardenafil may also provoke neurogenic-type rhinitis through their vasodilator properties, affecting the erectile tissue of nasal turbinates and causing nasal obstruction [5,41]. Rhinitis medicamentosa is characterized by rebound nasal congestion after excessive use of nasal decongestants. There are two groups of nasal decongestants responsible for this condition: sympathomimetic amines (caffeine, Benzedrine, amphetamine, mescaline, phenylpropanolamine, pseudoephedrine, phenylephrine, and ephedrine) and imidazolines (oxymetazoline, naphazoline, xylometazoline and clonidine). Sympathomimetic amines lead to presynaptic release of norepinephrine, which binds to postsynaptic a-receptors, resulting in vasoconstriction [5,43]. They are also b-receptor agonists and provoke vasodilation after the end of the a-effect [43]. Imidazolines are postsynaptic a2 agonists causing vasoconstriction and also decrease the endogenous norepinephrine via a negative feedback mechanism [43]. Apart from nasal decongestants, rhinitis medicamentosa may also be caused by cocaine. When nasal decongestants are used for prolonged periods, they may cause rebound nasal congestion and several histologic changes in nasal mucosa with more prominent epithelial oedema, squamous cell metaplasia, and goblet cell hyperplasia [44]. Treatment of drug-induced rhinitis consists of discontinuation of the responsible drug and instituting an intranasal corticosteroid or a combination of intranasal corticosteroid and intranasal antihistamine if the intranasal corticosteroid is not effective alone [31,41].

Idiopathic drug-induced rhinitis: it is caused by several drug classes, including b-blockers, angiotensin-converting enzyme inhibitors, calcium channel blockers, and antipsychotics. Many of these do not share similar pathophysiological mechanisms.

### 3.2. Rhinitis of Elderly

Rhinitis of the elderly or senile rhinitis is defined as clear watery anterior rhinorrhea, which is not associated with a specific triggering factor and is considered to be influenced by cholinergic hyperreactivity and age-related anatomic nasal changes. These induce collagen atrophy, degeneration of connective tissue and mucous glands, and weakening of the septal cartilage. It is hypothesized that such changes result in nasal congestion, drying, and decreased nasal flow, which may lead to local nasal mucosal atrophy, crusting, and symptoms of atrophic rhinitis [5].

### 3.3. Hormone-Induced Rhinitis

Hormone-induced rhinitis includes rhinitis of pregnancy and menstrual cycle-related rhinitis [5]. Rhinitis of pregnancy appears during the last two months of pregnancy and resolves within two weeks post-partum and is more common among smokers. Premenstrual rhinitis is related with pre-menstrual symptoms on a recurrent cyclic basis. The symptoms of hormone-induced rhinitis are thought to be mediated by elevated levels of estrogen, which provoke nasal congestion through vascular engorgement, although the exact mechanism has not been established [1,5]. There are also other endotypes involving beta-estradiol, which elevates the expression of H1 receptors of histamine on nasal epithelial and endothelial cells, causing eosinophilic migration and degranulation [5,45]. Other pathophysiologic mechanisms have been suggested including progesterone, insulin-like growth factor-1, prolactin, a variant of placental growth hormone and increased levels of human growth factor leading to vasodilation by the increase of the circulatory blood volume [1,45]. Another sub-phenotype of hormone-induced rhinitis associates the increased levels of human growth hormone with nasal mucosal hypertrophy [1,45]. Finally, thyroid disorders may be correlated with forms of hormonal rhinitis but without definite scientific evidence [45].

### 3.4. Gustatory Rhinitis

Gustatory rhinitis is a non-allergic, non-inflammatory type of rhinitis, characterized by the acute beginning of watery or mucoid rhinorrhea provoked by the ingestion of hot and spicy food. Hot chili peppers, red cayenne, Tabasco sauce, onion, chili, vinegar, red pepper, and mustard are among the most common instigators [46,47]. Capsaicin is an ingredient in all of the above and a triggering factor for the onset of gustatory rhinorrhea by stimulating afferent sensory nerves in the mucosa of the oropharyngeal cavity [47]. Four subgroups of gustatory rhinitis have been described: idiopathic, post-traumatic, post-surgical and gustatory rhinorrhea associated with cranial nerve neuropathy. All of these types may be unilateral or bilateral, except for idiopathic, which is always bilateral.

Gustatory rhinitis is not clearly associated with sex, age, or atopy. The peak prevalence is between 20 and 60 years [46,47]. The pathophysiologic mechanisms of gustatory rhinitis are still a focus of debate. According to one theory, ingestion of an offending food leads to gustatory rhinorrhea by stimulation of the trigeminal nerve endings which activates postganglionic, cholinergic, muscarinic, and parasympathetic fibers [46,47]. Another common pathophysiologic explanation is a hyperactive non-adrenergic, non-cholinergic neural system. Capsaicin activates, among others, the TRPV1 (transient receptor potential vanilloid receptors subtype 1), which is located in neuronal and non-neuronal cells in the nasal mucosa and oral epithelium [48]. Capsaicin causes secretory hyperfunction through activation of TRPV1 on submucosal glands and goblet cells. TRPV1 can also be found on sensory nerve C and Aδ fibers [48,49]. The stimulation of C fibers by capsaicin leads to local release of several neuropeptides, including substance P (SP) and calcitonin gene-related peptide (CGRP), and, as a result, increased vasodilation, secretions, and vascular permeability [46]. Through and following continuous exposure to capsaicin, there is a decrease in the available number of secretory peptides, and as result, desensitization occurs. For this reason, this desensitization comprises the basis for the therapeutic result of capsaicin in non-allergic rhinitis.

Sympathetic and parasympathetic nerve interaction may be another possible pathophysiological mechanism for gustatory rhinitis. Stimulation of sympathetic nerves causes the release of noradrenaline and neuropeptide Y (NPY), which works as a vasoconstrictor and neuromodulator for sensory and parasympathetic nerve function [50,51]. The reduction of NPY on parasympathetic activity could be the reason for the expression of gustatory rhinitis symptoms. The diagnosis of gustatory rhinitis is based on medical history and is essentially a diagnosis of exclusion. However, diagnosis may be established with the use of a relevant questionnaire such as that proposed by Waibel et al. [52].

Apart from that, there are also generally accepted stimulation tests for gustatory rhinorrhea, as proposed by Franceschini et al. [53]. Avoidance of the offending agents is of vital importance for the prevention of symptoms. Treatment includes intranasal anticholinergics prior to the consumption of responsible foods, or anticholinergic agent with an intranasal corticosteroid as a more effective choice [5,46]. Intranasal atropine and oxitropium bromide may also be used in gustatory rhinitis despite their side effects [5,46]. In addition, local capsaicin administration may lead to the degeneration of C fibers and long-term alleviation of gustatory rhinitis symptoms. Another option is botulinum toxin type A (BTXA) via injections in the mucosa of the middle and inferior turbinate, and nasal septum [46]. It is suggested to be effective in rhinorrhea through the blockage of acetylcholine release in cholinergic autonomic nerves and neuromuscular joint by binding to peripheral cholinergic receptors [52]. As a last resort for gustatory rhinitis management, Vidian nerve neurectomy or, rarely, posterior nasal nerve resection may be indicated [46].

### 3.5. Occupational Rhinitis

Occupational rhinitis is associated with nasal symptoms such as congestion, rhinorrhea, nasal itching, and sneezing, following exposure to specific agents in the workplace. It is considered as an umbrella term including allergic, non-allergic, irritant-induced, and neurogenic endotypes. For this reason, it is considered difficult to be classified with definite criteria in subgroups [54]. Occupational rhinitis also includes work-related rhinitis, in which rhinitis symptoms are developed in previously healthy individuals after the exposure to agents at work and work-exacerbated rhinitis, in which rhinitis symptoms occur in individuals with a previous history of rhinitis symptoms [55].

In allergic occupational rhinitis, the exposure to a specific agent leads to an IgE-mediated Th2 response of the immune system. However, some allergic causative agents provoke IgG-mediated immune responses [55]. Although non-allergic occupational rhinitis is not characterized by a specific immunologic pathophysiologic background, it involves mechanisms causing epithelial changes or damage and neurokinin release, which could play a role in pathogenesis [55,56]. Non-allergic occupational rhinitis includes rhinitis symptoms following single exposure to high amounts of a specific agent, which has been termed reactive upper airway dysfunction syndrome (RUDS) [55,56]. The irritants of occupational rhinitis consist of high-molecular-weight (HMG) agents that may be organic from plants, microorganisms, or animals, as well as low-molecular-weight (LMG) agents, which are inorganic. More reactions have been observed to occur after exposure to HMW than LMW irritants [55,56,57]. Occupational rhinitis is strongly correlated with occupational asthma. Most patients with occupational asthma suffer from occupational rhinitis, which also precedes the development of occupational asthma. Patients with occupational rhinitis are at a greater risk of asthma [54,55]. The worldwide prevalence of occupational rhinitis is estimated to be about 5% to 15% and two out of three people who suffer from occupational asthma also suffer from rhinitis [37]. However, the rate of subjects with occupational rhinitis who are going to develop occupational asthma remains uncertain [54]. The most prevalent hypothesis explaining this relation remains that of the united airway model, in which the expression of rhinitis and asthma is secondary to the same inflammatory events to the upper and lower airways [16,55]. Medical history is key for diagnosis, focusing on symptom type and duration, time of onset, alleviating and deteriorating factors, and clinical examination. The diagnostic approach includes SPT or RAST tests in order to assess sensitivity to specific agents [55]. However, LMW irritants typically do not induce an IgE-mediated response. Nasal provocation tests are considered as the gold standard for diagnosis [55,56]. Summarizing, a thorough history and clinical examination compatible with occupational rhinitis, and sensitization to a specific agent, as shown by a positive nasal provocation test, establish the diagnosis of occupational rhinitis. Primary prevention strategies with avoidance of exposure to irritant agents seem to be of vital importance for the management of occupational rhinitis. Secondary prevention strategies such as symptom questionnaires and immunological assays for evaluation of the sensitization may be used. Monitoring of patients with occupational rhinitis in order to prevent the development of occupational asthma should be strongly considered [55]. Medical treatment of occupational rhinitis includes intranasal corticosteroids, which may alleviate nasal symptoms. Immunotherapy is generally not indicated as an option for treatment [55].

### 3.6. Atrophic Rhinitis

Atrophic rhinitis is classified as primary and secondary. Both groups are characterized by crusting, obstruction, halitosis, purulent discharge, dryness, and epistaxis [1,5]. Nasal mucosa and glandular atrophy are prominent in primary atrophic rhinitis, which mainly affects people living in dry, warm climate areas [5]. The pathophysiological mechanisms are not well understood. Nasal dryness and a lack of mucosal secretions assist the bacterial growth and bacterial mucosal colonization (most common bacteria: Klebsiella ozaenae, Staphylococcus aureus, Proteus mirabilis, *Escherichia* coli). The opposite pathway may also be true [58]. The secondary subtype is caused by surgical removal of mucus secreting tissue, trauma, or granulomatous diseases, and is characterized by fetor, crusting, and nasal congestion [58]. The diagnosis of atrophic rhinitis is based on clinical suspicion and exclusion of autoimmune granulomatous diseases, as well as other causes of atrophic rhinitis such as tuberculosis, scleroma, syphilis, and leprosy [59].

Treatment of atrophic rhinitis is mainly conservative and may include nasal irrigations; glucose or glycerin use, which may inhibit infection by bacteria and other saprophytic organisms, and also promote the growth of nasal flora and improve nasal vascularity; and paraffin nose drops, which lubricate nasal mucosa and remove crusts [59]. Other options of medical management include chloramphenicol-streptomycin drops; nemicetene antiozaena solution, which contains chloramphenicol estradiol diproprionate, vitamin D2, and propylene glycol; acetylcholine with or without pilocarpine; vasodilators; and antibiotics following evaluation for infectious etiology [59]. Decongestants and antihistamines are strongly contraindicated. A rather novel treatment option that is gaining ground is placental extract submucosal injections. The extract is injected in each nasal cavity every week for a period of 24 weeks and has angiogenic and mitogenic activity. However, even if relief of nasal symptoms is accomplished, relapse following the cessation of treatment may be noted [59,60].

### 3.7. Idiopathic Rhinitis

The most common type of NAR is considered to be idiopathic rhinitis or vasomotor rhinitis. This type of rhinitis consists of approximately 71% of non-allergic rhinitis, with a worldwide prevalence of 320-million people and without a clear correlation with the rates of comorbid asthma [61,62] Idiopathic rhinitis is associated with symptoms which are not related to allergic infectious triggers, without a clear etiology [1,5]. The diagnosis is usually made following exclusion of AR. It seems that the prominent pathophysiological mechanism is based on flawed neurogenic activity since there is no correlation with systemic allergic diseases, structural defects, or distinct cellular inflammatory reactions.

Noxious odorants, chemical irritants, cleaning agents, and changes in environmental status such as temperature, humidity, and barometric pressure are considered to be triggering factors for the manifestation of idiopathic rhinitis. Several molecular pathways have been found to be involved; one of them is characterized by tachykinin release and the inhibition of mediators of the sympathetic system, leading to an elevated parasympathetic response [62]. This mechanism is not strongly supported by evidence. It is rather supported that at least some types of idiopathic rhinitis represent a malfunction of the non-adrenergic, non-cholinergic, or peptidergic neural system [1,5]. Inflammatory neuropeptides activate nasal peptidergic neurons that influence blood vascularity and secretory activity of mucous glands of the nasal cavity [5]. The peptidergic and especially C fibers are activated by TRP (transparent response potential calcium ion channels. The TRPs are activated after the connection with specific ligaments. Those ligaments are affected by chemical irritants, changes in temperature or mechanical–osmotic pressure [5].

Capsaicin is a ligand for transient potential receptor vanilloid 1 (TPRV1), which is activated by hot temperature [5,63]. Recurrent exposure to capsaicin may desensitize the TRPV1 receptor and capsaicin is considered a promising option of medical therapy. Similar calcium channels to TRPV1, transient potential receptor ankyrin 1 (TRPA1), and transient potential receptor melastin 8 (TPRM8) are activated by cold, dry air and also release capsaicin [63]. All of these pathophysiological pathways of idiopathic or neurogenic rhinitis are an umbrella that covers other entities such as gustatory rhinitis, occupational rhinitis, etc. Increased nasal responsiveness to non-specific physical and chemical triggers in inspired air is defined as nasal hyperreactivity. It is well established that apart from capsaicin anticholinergic drugs are accepted as to response of patients with idiopathic rhinitis.

## 4. Infectious Rhinitis

Nose and sinuses share common vascular and anatomic pathways, a fact that explains why rhinitis coexists with sinusitis. Acute viral rhinitis is the most common form of upper respiratory infection and is usually due to viral rather than bacterial agents [1,5]. Common causes of viral rhinitis include rhinovirus, coronavirus, adenovirus, influenza virus, parainfluenza virus, respiratory syncytial virus, and enterovirus [5,64]. These viruses provoke damages in tight junctions among epithelial cells, disrupt their membranes, invade the epithelial cells, and dominate host cell metabolic activity, using it for their development and causing host cell destruction and death. Usually, the symptoms of infectious rhinitis are self-limited and there is no need for medical therapy as the initial approach of the disease [64]. Antibiotic administration is not indicated for viral rhinitis, unless there is bacterial superinfection [1,5]. There are antiviral molecules such as interferon alpha (INF-α) that are proved to be effective in acute viral rhinitis by shortening the duration and severity of symptoms [64].

## 5. Conclusions

It is acceptable that rhinitis phenotypes are well defined clinically but the underlying pathophysiologic mechanisms of rhinitis are in many cases not adequately understood. Further investigation may result in well-targeted therapies, which, in turn, will offer superior symptom relief, control associated comorbidities, and decreased economic burden related to this highly prevalent condition.

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
