# Peer review of "Review of Rhinitis: Classification, Types, Pathophysiology"

_jcm, 2021, doi:10.3390/jcm10143183_

Round 1

Reviewer 1 Report

【General comments】

This manuscript is a review about the classification, types, and pathophysiology of rhinitis.

This is a very well-described review.

However, there are some improvements that should be made. 

【Specific comments】

(1)Chapter 3.5. Occupational Rhinitis Page7, Line8-9

How much is the correlation between occupational rhinitis and occupational asthma?

Please let me have a comment .

(2)Chapter 3.7. Idiopathic Rhinitis

It would be useful to include this information about vasomotor rhinitis.

I hope that my comments are useful for the improvement of this manuscript.

Author Response

We would like to thank you for the useful comments.

1st Reviewer

Point 1: How much is the correlation between occupational rhinitis and occupational asthma? Please let me have a comment .

Response 1: Most patients with occupational asthma suffer from occupational rhinitis, which also precedes the development of occupational asthma. Patients with occupational rhinitis are at greater risk of asthma [21,59]. The worldwide prevalence of occupational rhinitis is estimated about 5-15% and 2 of three people who suffer from occupational asthma also suffer from rhinitis [65]. However, the rate of subjects with occupational rhinitis who are going to develop occupational asthma remains uncertain [59].  Chapter 3.5, Page 7, Line 329-335

Point 2: It would be useful to include this information about vasomotor rhinitis.

Response 2: The most common type of Non-Allergic Rhinitis is considered to be idiopathic rhinitis or vasomotor rhinitis. This type of rhinitis consists approximately 71% of non allergic rhinitis conditions, with a worldwide prevalence of 320 million [67]. Idiopathic rhinitis is associated with symptoms which are not related to allergic infectious triggers, without a clear etiology [1,5]. Chapter 3.7, page 8, line 378-382

Reviewer 2 Report

This review of rhinitis is well written.

It is easy to understand and written in detail.

But, I have some comments.

In page 3 line 99. Authors explain about "RAST". Although the term "RAST" is still used conventionally, radioisotope is rarely used now. It is better to describe about new method such as Immuno CAP etc.

Comprehensive examination (e.g. MAST) should be explained. 

Additional general comments In this paper rhinitis are divided into two groups. (Allergic and Non allergic) Authors write in detail about minor rhinitis. But I think that the main rhinitis is allergic rhinitis and infective rhinitis. It should be described about infectious rhinitis.

It does not seem to be good to classify occupational rhinitis in non-allergic rhinitis. As writers recognize, it includes both allergic type and nonallergic type. I think it should be discussed about classification. It is difficult to classify it in allergic and non-allergic definitely.

Additional minor comment In page 3 line 126 Authors describe that Omalizumab has not been approved for treatment of AR. However it is not correct. At least, Omalizumab is approved for treatment for severe seasonal allergic rhinitis in Japan. Please update about Omalizumab.

Author Response

Point 1: Authors explain about "RAST". Although the term "RAST" is still used conventionally, radioisotope is rarely used now. It is better to describe about new method such as Immuno CAP etc. Comprehensive examination (e.g. MAST) should be explained. 

Response 1: The diagnostic tests for allergic rhinitis are separated into in-vivo which are skin prick tests and in-vitro including radioallergosorbent test (RAST), multiple antigen simultaneous testing (MAST), fluoroallergosorbent test (FAST) and immunoassay capture test (ImmunoCAP). The immunoCAP system is an in-vitro test using 3 dimensional cellulose solid allergen phase in order to detect specific IgE to allergen components. It has been found that immunoCAP tests have similar sensitivity for house dust and lower sensitivity for pollen, dog dander and Candida comparatively to skin prick tests [62]. The MAST test uses no radioactive agents and allows simultaneous examination of multiple antigens. The MAST system provides similar information as the CAP [60,61,62]. However CAP system seems to have better sensitivity [61]. The FAST is a method which measures specific serum IgE by a chemical radiating method like MAST and was firstly used as an inhibition assay in order to determine cross-reaction activity between aeroallergens [62,63]. In comparison with MAST FAST requires less time to be analyzed and less serum quantity to be used [62,63]. Chapter 2, page 2 line 98, page 3 line 99-101, 103-104, 115-124

Point 2: It should be described about infectious rhinitis.

Response 2: Nose and sinuses share common vascular, neural and anatomic pathways, a fact that explains why rhinitis coexists with sinusitis. Acute viral rhinitis is the most common form of upper respiratory infection and usually is due to viral rather than bacterial agents [1,5]. Common causes of viral rhinitis include rhinovirus, coronavirus, adenovirus, influenza virus, parainfluenza virus, respiratory syncytial virus and enterovirus [5,64]. These viruses provoke damages in tight junctions among epithelial cells, disrupt their membranes, invade the epithelial cells and dominate host cell metabolic activity using it for their development and cause host cell destruction and death. Usually, the symptoms of infectious rhinitis are self-limited and there is no need for medical therapy as the initial approach of the disease [64]. Antibiotic administration is not indicated for viral rhinitis, unless there is a bacterial superinfection [1,5]. There are antiviral molecules such as INF-α that are proved to be effective in acute viral rhinitis by shortening the duration and the severity of symptoms [64]. Chapter 4, page 9, line 410--421

Point 3: It does not seem to be good to classify occupational rhinitis in non-allergic rhinitis. As writers recognize, it includes both allergic type and nonallergic type. I think it should be discussed about classification. It is difficult to classify it in allergic and non-allergic definitely.

Response 3: It is considered as an umbrella term including allergic, non allergic, irritant-induced and neurogenic endotypes. For this reason it is considered difficult to be classified with definite criteria in subgroups [59]. Chapter 3.5, page 7, line 311-313

 Point 4:  Authors describe that Omalizumab has not been approved for treatment of AR. However it is not correct. At least, Omalizumab is approved for treatment for severe seasonal allergic rhinitis in Japan. Please update about Omalizumab.

Response 4: Omalizumab is an anti-IgE hummanised monoclonal antibody which is approved for chronic urticaria and severe allergic asthma [26]. It has been found that it reduces nasal and ocular symptoms in AR but has not been approved for treatment of AR yet [1,2,9]. Specifically it was conducted a randomized placebo control, double blinded trial which has proved that omalizumab prevents and controls nasal and occular symptoms in moderate-severe seasonal allergic rhinitis to pollen in Japan [65]. However, it has not been approved for seasonal allergic rhinitis yet[65]. Chapter 2, page 3, line 143-145